# A Novel Color Image Encryption Algorithm Based on 5-D Hyperchaotic System and DNA Sequence

**DOI:** 10.3390/e24091270

**Published:** 2022-09-09

**Authors:** Xinyu Li, Jian Zeng, Qun Ding, Chunlei Fan

**Affiliations:** Electronic Engineering College, Heilongjiang University, Harbin 150080, China

**Keywords:** 5-D hyperchaotic system, color image encryption, bit-level permutation, DNA encoding

## Abstract

Nowadays, it is increasingly necessary to improve the encryption and secure transmission performance of images. Therefore, in this paper, a bit-level permutation algorithm based on hyper chaos is proposed, with a newly constructed 5-D hyperchaotic system combined with DNA sequence encryption to achieve bit-wide permutation of plaintexts. The proposed 5-D hyperchaotic system has good chaotic dynamics, combining hyperchaotic sequence with bit-level permutation to enhance the pseudo-randomness of the plaintext image. We adopt a scheme of decomposing the plaintext color image into three matrices of R, G, and B, and performing block operations on them. The block matrix was DNA encoded, operated, and decoded. The DNA operation was also determined by the hyperchaotic sequence, and finally generated a ciphertext image. The result of the various security analyses prove that the ciphertext images generated by the algorithm have good distribution characteristics, which can not only resist differential attacks, but also have the advantages of large cryptographic space.

## 1. Introduction

With the rapid development of the digital economy, the application of the Internet is becoming more and more widespread, which is making people’s lives change radically. Work and study are no longer confined to books, and there is a growing trend toward mobile portable devices. In today’s world of huge amounts of data, the field of application of digital images as a vehicle for information dissemination is constantly being expanded, and images have the natural advantage of being informative and easy to transmit. Therefore, the security of digital images has become a key to image processing technology [1,2]. The security of images can affect both national security and personal life. In the past, there was a basis for digital watermarking technology [3,4,5,6]. Thus, due to the initial value sensitivity and randomness of chaotic systems, many scholars have applied them to image encryption.

In recent years, the commonly used encryption algorithms are mainly AES and DES algorithms, which can be applied to image encryption technology in theory, but for digital images with large data volume, high redundancy and high correlation, the encryption efficiency using AES and DES is low. However, image encryption algorithms based on chaotic systems are easy to implement in both software and hardware, which makes chaotic image encryption increasingly researched and widely used by scholars. In 2016, Ref. [7] proposed a new one-dimensional discrete chaotic system, co-controlled it with a logistic map for the encryption algorithm, and used both chaotic systems to jointly construct an S-box for permutation and diffusion. Later, Cavusoglu et al. [8] took a 3-D chaotic system to create S-Box for image encryption. In 2019, Ref. [9] proposed a dual image encryption algorithm based on a spatio-temporal chaotic system, and the results showed good security. Subsequently, Alawida et al. [10] took an image encryption algorithm based on hybrid chaos and chaotic perturbation of pixels, which allows for a larger parameter range and optimal chaotic behavior by coupling two one-dimensional discrete chaotic systems. Recently, Ref. [11] proposed a new discrete chaotic system TM-DFSM, combining a Tent map and finite state machine, where ciphertext images are obtained by two rounds of permutation and diffusion, and the complexity and larger key space of the chaotic system is improved by TM-DFSM. Ref. [12] proposed a three-dimensional Rubik’s cube for permutation and a one-dimensional logistic for diffusion. Zheng et al. [13] proposed an improved two-dimensional logistic chaotic map for image encryption, using a combination of logistic and sinusoidal mappings to enhance the chaotic properties. In 2020, Ref. [14] proposed a new chaotic system that combines a two-dimensional Lorenz chaotic system with a logistic map that was used for image encryption. Subsequently, Ref. [15] used a 3-D chaotic map with prime modular for pixel alignment, and row diffusion and hill diffusion were used for pixel replacement. By contrast, most of the image encryption algorithms are designed on the basis of low-dimensional discrete chaotic systems, which lead to a small key space for the encryption algorithms and can easily perform exhaustive attacks on their algorithms to decrypt ciphertext images. The control parameters and state variables of a hyperchaotic system are more than those of a low-dimensional chaotic system. Moreover, the structure of the hyperchaotic system is more complex, and the dynamics of the generated chaotic sequence is even better, so the encryption performance based on the hyperchaotic system is more secure. Therefore, in this paper, by designing a new 5-D hyperchaotic system, and analyzing the characteristics of the new 5-D hyperchaotic system, the results show that the new 5-D hyperchaotic system has good chaotic characteristics, and can be used in chaotic image encryption.

Chaotic image encryption algorithms are usually based on scrambling and diffusion, which are the main techniques with the purpose of obscuring the statistical properties of plaintext images. The combination of scrambling and diffusion can yield statistically better ciphertext images. Nowadays, there are only three common types of scrambling algorithms: the first is row scrambling, column scrambling and cross scrambling of image matrices; then, the second one converts the image matrices into one-dimensional vectors for position scrambling; the last one uses the scrambling matrices to change the positions of pixel points. In 2017, Zhang et al. [16] put forward an MIE algorithm that is based on mixed image elements and permutation, which improves the encryption efficiency by comparing with the traditional Arnold permutation algorithm, but has weaker resistance to cropping. In 2019, Ref. [17] proposed a neural network-based simultaneous dislocation and diffusion encryption algorithm that performs the dislocation diffusion part simultaneously, and can resist attackers against a single dislocation and overcome the drawbacks of the classical dislocation diffusion structure. In 2020, Ref. [18] proposed an image encryption algorithm using chaos and a Mandelbrot set, where the choice of Arnold map is associated with plaintext, which can avoid brute force attacks. In 2021, Ref. [19] proposed a color image encryption algorithm based on Fisher-Yates permutation algorithm and DNA sequence operation. The color image was decomposed into three components, R, G and B, and was subjected to Fisher-Yates permutation operation; their experimental results proved that the algorithm has good robustness. Moreover, a permutation method based on chaotic Josephus perturbations was chosen in ref. [20], where the permutation and diffusion matrices were generated by a chaotic sequence. The combination of chaotic sequences and Josephus is used to improve randomness. In 2021, Ref. [21] divided the image into eight bit-planes, randomly divided into three parts using binary tree, flip scrambling and index scrambling, and diffusion operation by improving the GF(257) domain. However, the anti-differential performance was poor. Nevertheless, the single use of row disarrangement and column disarrangement reduces the ciphertext security, and using the plaintext attack method, only a certain amount of plaintext is required to restore half of the information in the ciphertext image. The chaos matrix is cyclic, and will restore the plaintext image after multiple chaotic iterations: thus, the security is extremely low. Although more and more chaotic image encryption algorithms are proposed, most of the schemes are weak against differential attacks. The key is that these encryption schemes do not achieve strong permutation: bit-level permutation has good encryption effect compared with pixel-level permutation, which only changes the pixel position, while bit-level permutation can change both position and size. In this paper, by designing a new bit-level permutation method, and reordering and diffusing pixel values, we show through our results that our proposed method has a good performance advantage.

DNA coding is used for image encryption by many scholars studying image encryption due to its biological self-encryption properties. Research has shown that DNA computing can simulate DNA biological operations to encrypt information, thus improving the security and efficiency of image encryption algorithms. At present, scholars have combined DNA coding and chaotic systems to propose some new image encryption algorithms. Thus, ref. [22] proposed a rule matrix for DNA encoding and decoding with 2D-LASM, where the images are encoded according to the generated DNA encoding rules, the plaintext DNA matrix is rank-swapped, and the final plaintext matrix is obtained by heterodyning the DNA matrix with a key matrix. In addition, ref. [18] combined Arnold mapping with DNA coding, that is, coding the R, G and B components separately, and finally performing DNA operations on the matrix generated by Arnold mapping and shifting it through the Mandelbrot dataset. In this paper, by dividing the pixel matrix into blocks, the block-based matrix performs DNA operations. The randomness of the encryption algorithm is further improved.

## 2. Related Work

Recent studies have shown that traditional chaotic mapping suffers from the phenomenon of cycles [23,24], i.e., an encryption algorithm that uses an Arnold chaotic mapping will revert to a plaintext image after repeated iterations, which reduces the security of the encryption algorithms. In this paper, an image encryption algorithm based on a hyperchaotic system is proposed. The ciphertext can be disrupted and diffused at the same time by bit-level permutation, and the DNA encoding, decoding and operation are performed on the R, G, B surfaces after permutation and diffusion. The initial value of the hyperchaotic system is determined by the information in the plaintext image, while the DNA encoding and decoding is determined by the hyperchaotic sequence. The chaotic matrix is generated by a one-dimensional discrete chaotic system, which performs a DNA operation with the ciphertext matrix; the result of the calculation is then subjected to rank index permutation. From the above operations, the final ciphertext image is obtained. The algorithm security verification results prove that the proposed algorithm in this paper has a uniform pixel distribution, low pixel correlation, high security and can effectively resist a range of attacks.

The rest of this paper is arranged as follows: In Section 3, a new 5-D hyperchaotic system and the corresponding bit-level dislocation rules are proposed; and the dynamics of the hyperchaotic system will be presented. Furthermore, hyperchaotic systems are designed as specific circuits to meet practical requirements. The specific steps of the encryption algorithm in this paper will be discussed in Section 4. Then, in Section 5, the experimental results are given and the security of the encryption algorithm is analyzed. Finally, this paper is concluded in Section 6.

## 3. Preliminaries

### 3.1. A New 5-D Hyperchaotic System

Hyperchaotic systems have been the focus of attention since they were first proposed [25]. Hyperchaotic systems have more complex dynamic characteristics than chaotic systems, and hyperchaotic systems have more control parameters. In this paper, a new 5-D hyperchaotic system is proposed, and the mathematics expression is defined as
(1)$$\left\{\begin{array}{l} \dot{x}=ay-bx+yz+cw^{2}+d\\ \dot{y}=a_{1}x-b_{1}y-xz-c_{1}u\\ \dot{z}=d_{1}z+xy\\ \dot{w}=hyz+\dot{j}w\\ \dot{u}=ky \end{array}\right.$$
where x, y, z, w, u are the state variables of the hyperchaotic system, and a, b, c, d, $a_{1}$, $b_{1}$, $c_{1}$, $d_{1}$, h, j, k are the control parameters of the hyperchaotic system. When a=23.8, b=14.2, c=0.35, d=0.2, $a_{1}=30.9$, $b_{1}=-4.39$, $c_{1}=1.07$, $d_{1}=-1$, h=−0.38, j=−10.6, k=1, the system behaves in a hyperchaotic state. After introducing each of the above parameters into the hyperchaotic system, the motion characteristics of chaos can be directly observed in the phase diagram, and the phase diagram of the hyperchaotic system is represented in Figure 1.

The Lyapunov exponent and the bifurcation diagram are analyzed to determine whether the chaotic system has the important characteristics contained in the chaotic behavior. Likewise, the above parameters are brought into the system and the initial values are chosen as [2, 1, 25, 1, 1]. Figure 2 shows the evolution curves of the five Lyapunov exponents as a function of the parameter *k*: when the parameter *k* = 1, the Lyapunov exponents of the 5-D hyperchaotic system are calculated as $LE_{1}=2.635$, $LE_{2}=0.138$, $LE_{3}=0$, $LE_{4}=-10.371$ and $LE_{5}=-13.535$.

The Lyapunov exponent has two positive numbers, one 0, and two negative numbers. Therefore, the proposed chaotic system is a hyperchaotic system. The transition of a system from regularity to irregularity is characterized by the bifurcation diagram of a chaotic system. By keeping the remaining control parameters constant and choosing the value of k as [0,4), the bifurcation diagram of the hyperchaotic system is shown in Figure 3.

The complexity of a chaotic system refers to the degree to which a chaotic sequence is close to a random sequence by using a correlation algorithm. The larger the complexity value, the closer the sequence is to a random sequence. In this paper, the Spectral Entropy (SE) is used to test the discrete chaotic sequence, and the result is shown in Figure 4. It can be seen from the figure that the complexity value of the chaotic sequence is very large, and the complexity of the chaotic system is very high.

### 3.2. Dissipativity of the Chaotic System

The dissipativity of the hyperchaotic system is calculated as
(2)$$\nabla V=\frac{\partial \dot{x}}{\partial x}+\frac{\partial \dot{y}}{\partial y}+\frac{\partial \dot{z}}{\partial z}+\frac{\partial \dot{w}}{\partial w}+\frac{\partial \dot{u}}{\partial u}=-b-b_{1}+d_{1}+j+0$$
where b=14.2, $b_{1}=-4.39$, $d_{1}=-1$ and j=−10.6.

Hence, based on Equation (2), the hyperchaotic system proposed by this paper is a dissipative system which converges at an exponential rate $e^{-21.41t}$. For the volume element, $V_{0}$ converges to $V_{0}e^{-21.41t}$ at time *t*, and as t→∞, $V_{0}\to 0$. Since the phase point trajectory curves of the system will all be confined to a set whose volume is 0, the designed hyperchaotic system in this paper with singular attractors is effectively confirmed.

### 3.3. Equilibrium Point Analysis of the Chaotic System

The mathematical formula for the equilibrium points of the hyperchaotic system can be described as
(3)$$\left\{\begin{array}{l} ay-bx+yz+cw^{2}+d=0\\ a_{1}x-b_{1}y-xz-c_{1}u=0\\ d_{1}z+xy=0\\ hyz+jw=0\\ ky=0 \end{array}\right.$$

According to Equation (3), the unique equilibrium point O (0.0141, 0, 0, 0, 0.4067) of the hyperchaotic system can be obtained. Thus, the Jacobi matrix is expressed as
(4)$$J=\left[\begin{array}{ccccc} -b & a+z & y & 2cw & 0\\ a_{1} & -b_{1} & -x & 0 & -c_{1}\\ y & x & d_{1} & 0 & 0\\ 0 & hz & hy & j & 0\\ 0 & k & 0 & 0 & 0 \end{array}\right]$$

The eigenvalues of the matrix are 23.7325, 0.0191, −1, −10.6 and−33.5616. When one of the real parts of all the eigenvalues of the Jacobi matrix is positive, this corresponds to an unstable equilibrium state. It follows that the system has two eigenvalues greater than 0. Therefore, the point O is an unstable point where chaotic attractors can be formed.

### 3.4. A Simple Chaotic Pseudo-Random Number Generator

Most chaotic sequence quantization methods are mainly composed of basic operations such as modulo, rounding, and expansion. While these methods improve the randomness of chaotic sequences, they also have the disadvantage of a large amount of computation. Therefore, in this paper, we proposed a quantization method with XOR transformation, which is not only simple to operate, but also enhances the randomness of chaotic sequences. Notably, the chaotic pseudo-random sequence is denoted as $x_{n}$, and this quantization method is introduced as
$$\begin{array}{ll} \mathrm{Step}\;1.\; & d(i)=x(n+1)-x(n)\\ \mathrm{Step}\;2.\; & s_{1}(i)=\left\{\begin{array}{ll} 1 & d(i)>0\\ 0 & d(i)<0 \end{array}\right.\\ \mathrm{Step}\;3.\; & s_{2}(i)=\mathrm{mod}(floor(x(n)*2^{8}),2)\\ \mathrm{Step}\;4.\; & s(i)=bitxor(s_{1}(i),s_{2}(i)) \end{array}$$
where d(i) is the latter term minus the former term in the chaotic sequence. The result of $s_{1}(i)$ depends on the value of d(i). The result of $s_{2}(i)$ is obtained by modulo and rounding operations on $x_{n}$, XOR $s_{1}(i)$ and $s_{2}(i)$ to get s(i). s(i) is the quantized chaotic sequence. Based on the above operation steps, the quantized results s(i) were run through version 2.1.2 of the NIST SP-800-22 test [26]. The experiment results are listed in Table 1. The results have shown that the chaotic sequences quantized by XOR have excellent pseudo-random performance.

### 3.5. Cosine-Transform-Based Chaotic System

The digitization of a chaotic system will lead to the degradation of dynamic characteristics [27], in order to get the chaotic map to have more complex dynamic behavior. Based on cosine transform, Hua et al. [28] proposed a chaotic system which can produce complex dynamic behavior with the purpose of effectively resisting the dynamic degradation under the influence of limited accuracy. The mathematical expression of the chaotic map is described as
(5)$$x_{i+1}=\cos\left(\pi \left(4rx_{i}\left(1-x_{i}\right)+(1-r)\sin\left(\pi x_{i}\right)-0.5\right)\right)$$

Figure 5 shows the bifurcation diagrams of the cosine and logistic maps with varying parameter *r*. From the bifurcation diagram, it is obvious that the dynamic behavior of the cosine chaotic map is more complex.

### 3.6. Chaotic Circuit Simulation

In this paper, we use the general idea of modular circuit design and the specific chaotic circuit is shown in the Appendix A. The input and output voltages were put in the range of ±10 to ±50 V, which exactly corresponds to the range of values for each state variable. The time domain waveforms and phase diagrams of the chaotic system were obtained through Multisim hardware circuit simulation, and the results were consistent with the numerical simulation results of MATLAB, which established that the chaotic circuit designed and the simulation of Multisim software are fully practicable. The attractor plots from the Multisim simulation can be found in Figure 6. The purpose of the circuit simulation is to provide the basis for the subsequent hardware implementation.

### 3.7. Bit-Level Permutation

This paper used a combination of bit-level permutation and chaotic sequences. While the randomness is further improved, it can also achieve the purpose of diffusion. First, the size of the image is *M* × *N*, the chaotic sequence ***a*** is sorted with a size of 1 × 4 *MN* in ascending order, and the index vector ***b*** is generated by sorting. Next, the first to fourth columns of the high 4 bit-plane are joined into a 1 × 4 *MN* one-dimensional 0–1 vector, which is denoted as vector ***c***, and this is rearranged according to the index position of the vector ***b***. The aligned one-dimensional vector thus generated is denoted as vector ***d***. Finally, the vector ***d*** is rearranged into an *MN* × 4 two-dimensional matrix. The whole process can be described in Figure 7.

Since the chaotic sequences are determined by the plaintexts, the plaintext relevance of the algorithm is greatly enhanced. Correspondingly, choosing a different plaintext means the scrambling position will also be different. Therefore, three different chaotic sequences will be used for the R, G and B bits of the image. Accordingly, the randomness and security of this algorithm will be greatly enhanced by the combination of bit-level permutation and chaotic sequences.

### 3.8. DNA Coding

DNA in biology consists of base pairs. Base pairs are made up of A (adenine), T (thymine), G (guanine), and C (cytosine). Accordingly, DNA coding is borrowed from DNA in biology, where DNA coding encodes binary 00, 11, 10 and 01 as the corresponding base pairs A, T, G and C. According to Watson–Crick’s rule of complementarity, out of 4!=24 codes, only 8 codes fit the rule. The coding rules are shown in Table 2.

When an encryption algorithm takes one of the encoding rules to encode while another decoding rule is taken, it will effectively encrypt the pixel values. DNA can be encoded and then subjected to DNA operations, which include addition, subtraction, XOR and XNOR operations. For example, we adopted rule 1 from the table to encode the pixel value 228, while obtaining a DNA sequence with a value of TGCA, respectively, and then, decoded it with rule 4 to obtain a pixel value of 27. The whole process proved to be effective in protecting the plaintext. Table 3, Table 4 and Table 5 list the specific rules of the above DNA operations.

## 4. Color Image Encryption Algorithm

The process of the encryption algorithm proposed in this paper can be illustrated in Figure 8.

In order to explain the encryption algorithm proposed in this paper, we use a color image of size *M* × *N* as a plaintext image to demonstrate the flow of the encryption algorithm. Correspondingly, the decryption algorithm is the inverse of the encryption algorithm. The process of encryption algorithm is as follows:

**Step 1.** First, decompose the color image into three planes, R, G, and B, which are represented as red, green, and blue, which correspond to the constituent elements of a color image. As shown in Equation (6), the three decomposed matrices are called $Y_{1},Y_{2},Y_{3}$.
(6)$$\left\{\begin{array}{l} Y_{1}=Y(:,:,1)\\ Y_{2}=Y(:,:,2)\\ Y_{3}=Y(:,:,3) \end{array}\right.\;$$

**Step 2**. Next, in order to increase the adaptability of the encryption algorithm, it needs to be filled with 0 before the block so that the three matrices can be divided into blocks of the same size; therefore, the size of the matrix must satisfy the following conditions:(7)$$\left\{\begin{array}{l} \mathrm{mod}(M,t)=0\\ \mathrm{mod}(N,t)=0 \end{array}\right.$$
where *t* is the size of the block, 0 makes both *M* and *N* divisible by t, and the matrices $Y_{1},Y_{2},Y_{3}$ can be decomposed into t×t sized blocks.

**Step 3**. Then, the *M* × *N* decimal matrices $Y_{1},Y_{2},Y_{3}$ are transformed into binary 8 × *MN* matrices, and the upper 4-bit plane 4 × *MN* matrix is rearranged using bit level permutation. Equally, the lower 4-bit planes are combined from top to bottom for odd-numbered columns, and from bottom to top for even-numbered columns. The scrambled matrices are re-reduced to *M* × *N* decimal matrices, $P_{1},P_{2},P_{3}$.

**Step 4.** The cosine map function is iterated *M* × *N* + 3000 times, discarding the first 3000 times to obtain better randomness. In order to avoid the degradation of the chaotic sequence with finite accuracy, a small perturbation of the initial value is performed every 3000 iterations, the initial value after the perturbation is
(8)$$x_{i+1}=x_{i}+0.002*\sin(x_{i})\;$$
where $\left\{S_{i}\right\}$ is the generated chaotic sequence, whereas the initial values $x_{0}$ and r are denoted as one of the keys. Furthermore, the chaotic sequence $\left\{S_{i}\right\}$ is transformed into a decimal number from 0–255, which can be converted into an *M* × *N* matrix $P_{4}$, and we take the Reshape function, in which the $x_{0}$ can be expressed as
(9)$$x_{0}=\frac{\mathrm{sum}(Y_{1}(:))+\mathrm{sum}(Y_{2}(:))+\mathrm{sum}(Y_{3}(:))}{255*M*N*3}$$
According to Equation (9), we can obtain the result that $x_{0}$ is the average of the pixel greyscale of $Y_{1},Y_{2},Y_{3}$.

**Step 5.** Iterate the five-dimensional hyperchaotic system, the initial value of the system is selected by Equation (10), and the five chaotic sequences $\left\{X_{i}\right\}$, $\left\{Y_{i}\right\}$, $\left\{Z_{i}\right\}$, $\left\{W_{i}\right\}$, $\left\{U_{i}\right\}$ are obtained
(10)$$\left\{\begin{array}{l} X(0)=\mathrm{sum}\left(\mathrm{sum}\left(\mathrm{bitand}\left(P_{1},129\right)\right)\right)/(129\times M\times N)\\ Y(0)=\mathrm{sum}\left(\mathrm{sum}\left(\mathrm{bitand}\left(P_{2},66\right)\right)\right)/(66\times M\times N)\\ Z(0)=\mathrm{sum}\left(\mathrm{sum}\left(\mathrm{bitand}\left(P_{3},36\right)\right)\right)/(36\times M\times N)\\ \begin{array}{l} W(0)=\mathrm{sum}\left(\mathrm{sum}\left(\mathrm{bitand}\left(P_{1},24\right)\right)\right)/(24\times M\times N)\\ U(0)=\mathrm{sum}\left(\mathrm{sum}\left(\mathrm{bitand}\left(P_{2},17\right)\right)\right)/(17\times M\times N) \end{array} \end{array}\right.$$
where five chaotic sequences $\left\{X_{i}\right\}$, $\left\{Y_{i}\right\}$, $\left\{Z_{i}\right\}$, $\left\{W_{i}\right\}$, $\left\{U_{i}\right\}$ are done AND operation of 129, and the average of the first and eighth planes of $P_{1}$ can be obtained. Thus, we convert $\left\{X_{i}\right\}$ into a random integer from 1 to 8 to determine the DNA encoding rules for the sub-blocks in the same position of $P_{1},P_{2},P_{3}$. Similarly, transforming $\left\{Y_{i}\right\}$ into a random integer from 1 to 8 determines the coding rules for $P_{4}$; and transforming $\left\{Z_{i}\right\}$ into random integers from 1 to 4 will determine the DNA operation rules for $P_{1},P_{2},P_{3}$ and $P_{4}$. In addition, the DNA decoding rules that are determined as above after the DNA manipulation are converted to random integers $\left\{W_{i}\right\}$ from 1 to 8, and alongside $\left\{U_{i}\right\}$ are used to form the index matrix required for bit-level scrambling.

**Step 6.** Divide $P_{1}$,$P_{2}$, and $P_{3}$ into blocks, and the size of each sub-block is t×t. In order to improve the efficiency of the encryption algorithm, the same position sub-blocks of $P_{1}$, $P_{2}$ and $P_{3}$ are used for the same DNA encoding method, DNA operation and DNA decoding. The end of the operation is transformed into a decimal matrix.

**Step 7.** The cosine map function is adopted to generate two chaotic sequences $\left\{S_{x}\right\}$ and $\left\{S_{y}\right\}$ with sizes *M* and *N*. The selection of the initial value is determined by Equation (11), and the index matrix is obtained by descending order, taking $U_{x}$, $U_{y}$ sequence values and their corresponding indices as row and column exchange coordinates, and Equation (12) is used to perform row and column permutation to improve the cropping resistance of the image.
(11)$$\left\{\begin{array}{l} x_{01}=\frac{\mathrm{sum}\left(Y_{1}(:)\right)+\mathrm{sum}\left(Y_{2}(:)\right)}{255\times M\times N\times 2}\\ x_{02}=\frac{\mathrm{sum}\left(Y_{2}(:)\right)+\mathrm{sum}\left(Y_{3}(:)\right)}{255\times M\times N\times 2} \end{array}\right.$$
(12)$$\left\{\begin{array}{l} U_{x}=\mathrm{sort}\left(S_{x},\;‘\mathrm{descend}’\right)\\ U_{y}=\mathrm{sort}\left(S_{y},‘\mathrm{descend}’\right) \end{array}\right.$$

**Step 8.** Finally, based on the above steps, combine the encrypted three two-dimensional matrices into a three-dimensional matrix to obtain the final ciphertext image.

## 5. Image Algorithm Security Analysis

### 5.1. Encryption and Decryption Results

In order to verify the effectiveness of the algorithm, we conducted experiments on Lena, Baboon and Pepper of size 512 × 512 under a Windows 10 environment using MATLAB 2018b. The experimental results are shown in Figure 9, from which it can be established that the original image and the decrypted image have no distortion or data loss, while the cipher image has lost all features of the plaintext image: thus, the experimental results prove that the algorithm has better security.

### 5.2. Keyspace Analysis

The ability of an encryption algorithm to resist exhaustive attacks is reflected by the size of the key space. That is to say, if the key space is larger than $2^{128}$, this means that it is better resistant to exhaustive attacks. The key in the encryption process of this paper includes the control and initial values of the cosine map, as well as the initial values of the 5-D hyperchaotic system. Therefore, when computational precision is 32, the key space is roughly equal to $2^{13\times 32}$. According to the result above, the key space of the algorithm proposed in this paper is large enough to resist exhaustive attack.

### 5.3. Key Sensitivity Analysis

Key sensitivity is an important feature in evaluating the quality of an encryption algorithm, and a small change in key can produce extremely strong sensitivity. We changed the initial value $x_{0}$ of the cosine map from 0.5012 to 0.5012000000000001, and obtained the encrypted image shown in the Figure 10, which is proof that even with small changes, the encrypted image is completely different.

### 5.4. Correlation Analysis

As there is a correlation between pixels in the plaintext image, the encryption algorithm is used to reduce the correlation and the encrypted correlation should ideally be 0. The equations to calculate the correlation are given as
(13)$$r_{xy}=\frac{\left|\mathrm{Cov}(x,y)\right|}{\sqrt{D(x)\times D(y)}}$$
(14)$$\mathrm{Cov}(x,y)=\frac{1}{N}\sum_{i=1}^{N}\left(x_{i}-E(x)\right)\left(y_{i}-E(y)\right)$$
(15)$$E(x)=\frac{1}{N}\sum_{i=1}^{N}x_{i}$$
(16)$$D(x)=\frac{1}{N}\sum_{i=1}^{N}{\left(x_{i}-E(x)\right)}^{2}$$

Similarly, the detailed data for calculating the correlations for this algorithm are shown in Table 6. The correlation between the plaintext and ciphertext in each direction is represented separately in Figure 11. In conclusion, the encrypted pixel point correlation is low and the encryption algorithm proposed in this paper shows a high level of security to protect against statistical analysis of the ciphertext information by attackers.

### 5.5. Statistical Characterization

The histogram defines the gray level frequency of the image, and the RGB image histograms of plaintext and ciphertext are shown in Figure 12, which clearly shows the average amount of data in the ciphertext, indicating that the encrypted image masks all the original information. The grey level frequency of the image is defined by the histogram. The RGB image histogram for both plaintext and ciphertext is shown in Figure 11, which clearly shows the average amount of data in the ciphertext and indicates that the encrypted image masks all the original information.

### 5.6. Information Entropy

The uncertainty of the image information can be expressed by the information entropy of the image, which is calculated as
(17)$$H=-\sum_{i=0}^{L}p(i)\log_{2}p(i)$$
where the image gray level is denoted as *L*, and the probability of occurrence of gray value is denoted as p(i). In addition, the ideal value of *H* is 8. Lena 256 was evenly selected for the information entropy measure.

The comparison of the information entropy of this algorithm with other cryptographic algorithms is shown in Table 7. The ciphertext image information entropy of this algorithm is close to the ideal value, with high uncertainty and little visible information. Based on the above results, the algorithm proposed in this paper is proved to be highly secure.

### 5.7. Noise Attack

During image acquisition and transmission, the encrypted image will certainly be affected by some noise. Hence, the ability to resist certain noise interference is a measure of the cryptographic algorithm’s performance. In this paper, pepper noise of strength 0.05 and 0.1 is added to the encrypted image and decrypted with the correct key. The results are shown by Figure 13, which proved that the algorithm can still largely recover the plaintext image even though a certain amount of noise interference is added, which indicates that the algorithm is highly resistant to interference.

### 5.8. Anti-Crop Analysis of Ciphertext Images

Attackers can intercept and corrupt parts of the data in the ciphertext image during image transmission, and in general, the information will be very difficult to recover after loss. Corrupting the inter-pixel correlation can greatly improve the cropping resistance of the algorithm. In this paper, different levels of cropping attacks are applied to different locations of the ciphertext image, and the test results are shown in Figure 14, which confirming that the algorithm can still have some ability to recover the plaintext under cropping attacks, thus demonstrating the strong robustness of the algorithm.

### 5.9. Differential Attack

In our next step, we perform a differential attack on the encryption algorithm by making small changes to the plaintext, and the difference between the cipher text before and after the changes could obtained. The encryption algorithm is sensitive to changes in the plaintext, that is to say, a small change in the plaintext will cause a large change in the ciphertext. NPCR (Number of Pixel Change Rate) and UACI (Uniform Average Change Intensity) are used to encrypt images to measure the variation in the degree of difference between encrypted images. The ideal value for NPCR is 99.6094%; similarly, the ideal value for UACI is 33.4635%, which is calculated as
(18)$$NPCR=\frac{\sum_{i=1}^{M}\sum_{j=1}^{N}D(i,j)}{M\times N}\times 100\%$$
(19)$$UACI=\frac{\sum_{i=1}^{M}\sum_{j=1}^{N}\left|P_{1}(i,j)-P_{2}(i,j)\right|}{255\times M\times N}\times 100\%$$
where the relationship between $P_{1}(i,j)$ and $P_{2}(i,j)$ is
(20)$$D(i,j)=\left\{\begin{array}{ll} 0 & P_{1}(i,j)=P_{2}(i,j)\\ 1 & P_{1}(i,j)\neq P_{2}(i,j) \end{array}\right.$$

Table 8 and Table 9 list the various comparisons between the encryption algorithm proposed in this paper and those introduced in other papers. Based on Table 9, it is shown that the NPCR and UACI of the encryption algorithm proposed in this paper are closer to the ideal values than many other encryption algorithms, and that the proposed algorithm can effectively resist the anti-contrast attack.

## 6. Discussion

In this paper, a new 5-D hyperchaotic system is proposed, which has not only a large Lyapunov exponent but also other good properties. The larger the Lyapunov exponent, the faster the divergence of adjacent trajectories of the system; and this is the source of the sensitive dependence of chaos on initial conditions. In addition, a combination of bit-level permutation and DNA sequences were used to encrypt the color images. The plaintext color image is decomposed into three matrices, R, G and B, and a chunking operation is performed on these matrices, followed by DNA encoding, computation, and decoding. Simultaneously, the DNA operations are based on hyperchaotic sequences. The generated cryptographic images have been tested in various security tests and the experimental results have proved the excellent performance of the algorithm proposed in this paper. Finally, circuit simulations of chaos are performed, which provide the basis for future practical applications of color image encryption algorithms.

## Figures and Tables

**Figure 1 entropy-24-01270-f001:**
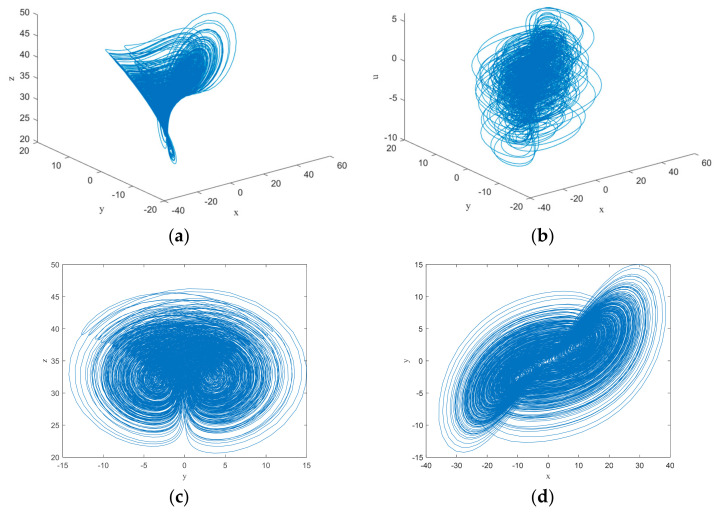
Phase diagrams of the hyperchaotic system: (**a**) *x*-*y*-*z* plane (**b**) *x*-*y*-*u* plane (**c**) *y*-*z* plane (**d**) *x*-*y* plane (**e**) *x*-*z* plane and (**f**) *w*-*u* plane.

**Figure 2 entropy-24-01270-f002:**
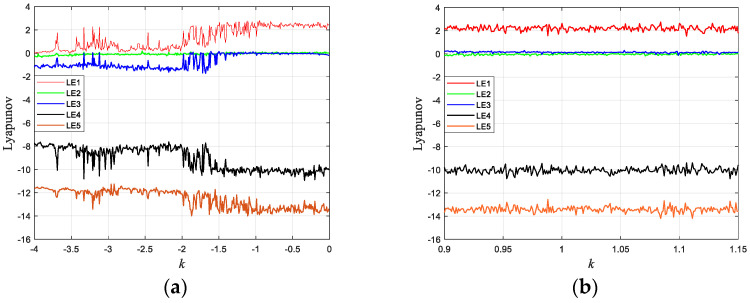
Lyapunov exponent spectrum: (**a**) k∈[4,0); (**b**) k∈[0.9,1.1].

**Figure 3 entropy-24-01270-f003:**
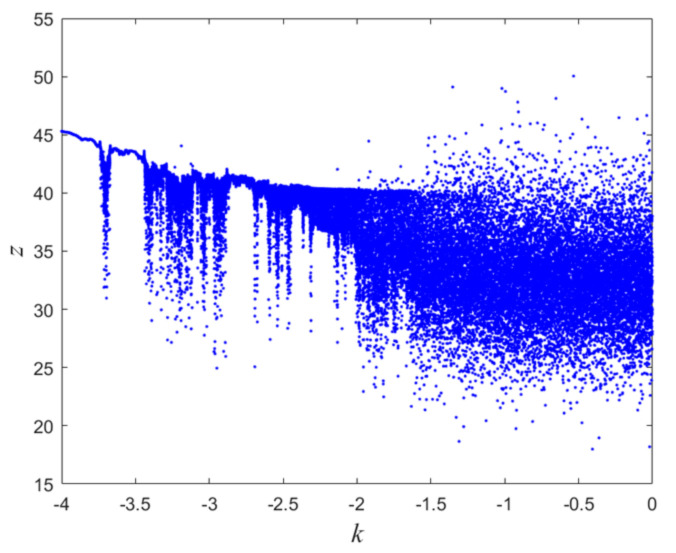
Bifurcation diagram with k∈[4,0).

**Figure 4 entropy-24-01270-f004:**
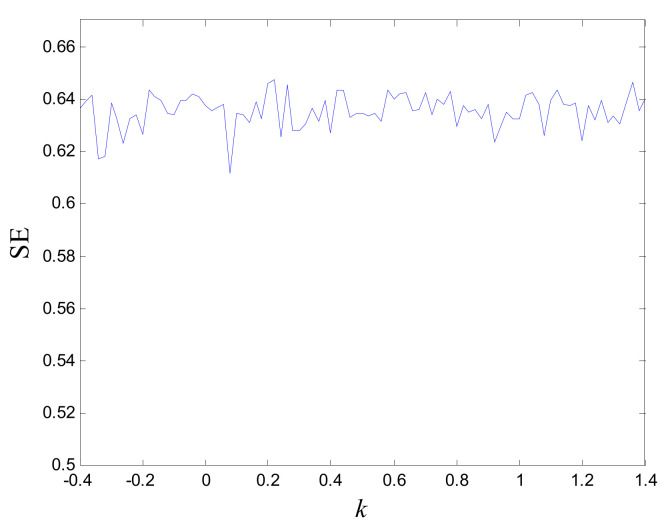
SE of the discrete chaotic sequence.

**Figure 5 entropy-24-01270-f005:**
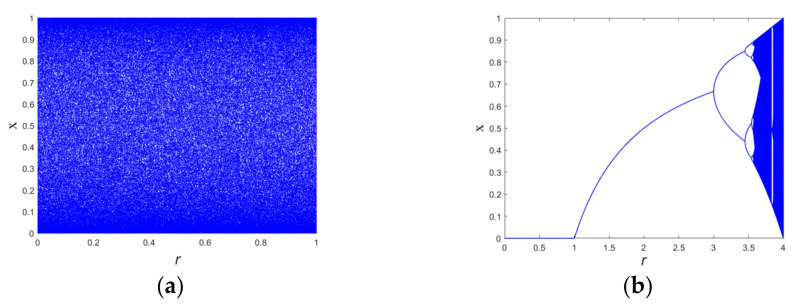
Bifurcation diagrams of different chaotic maps: (**a**) cosine map, and (**b**) logistic map.

**Figure 6 entropy-24-01270-f006:**
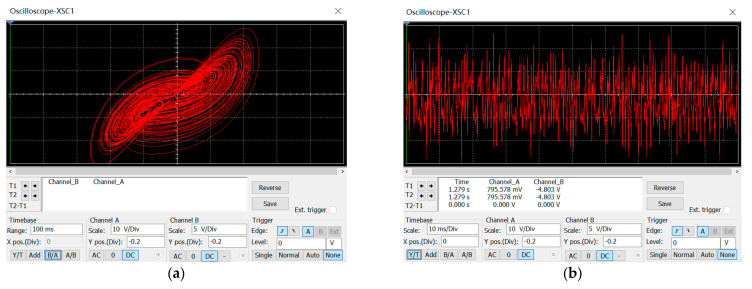
Multisim hardware circuit simulation: (**a**) circuit attractor simulation, and (**b**) circuit chaotic sequence simulation.

**Figure 7 entropy-24-01270-f007:**
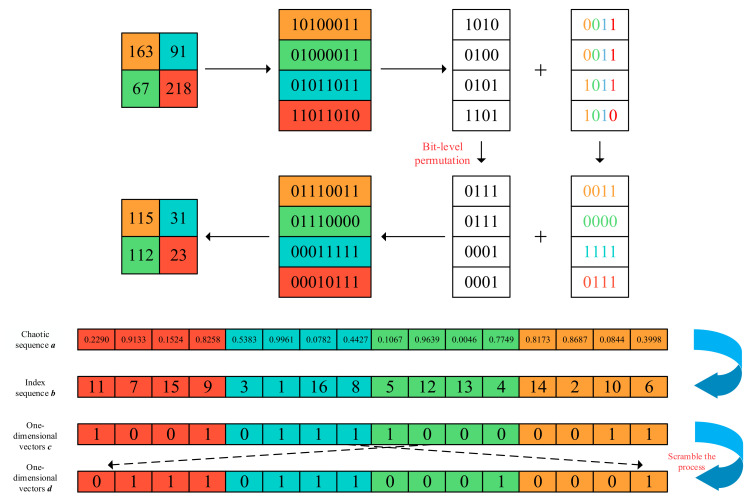
Flow chart of Bit-level permutation.

**Figure 8 entropy-24-01270-f008:**
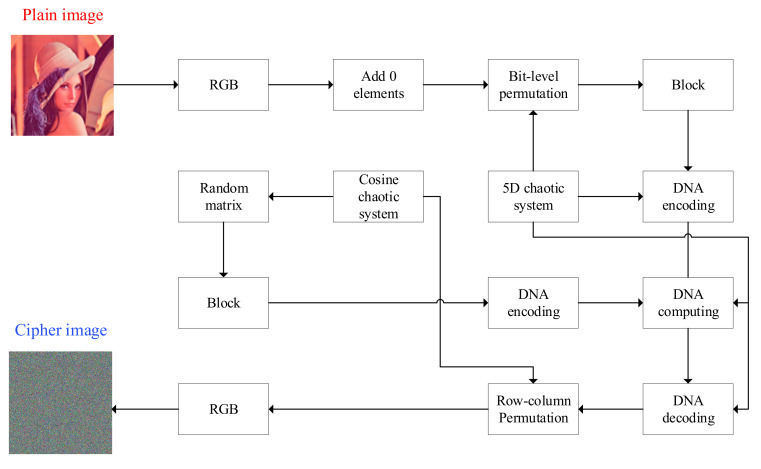
The overall frame structure of the proposed image encryption algorithm.

**Figure 9 entropy-24-01270-f009:**
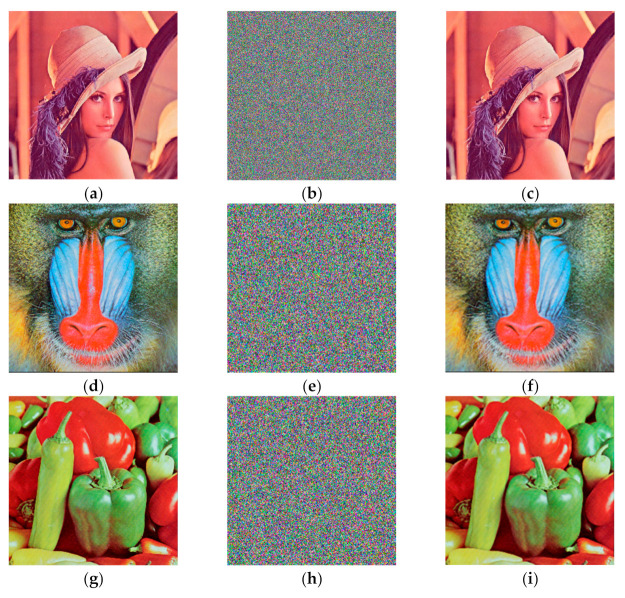
Encryption and decryption results. (**a**) Lena original image; (**b**) ciphered image of Lena; (**c**) decrypted image of Lena; (**d**) Baboon original image; (**e**) ciphered image of Baboon; (**f**) decrypted image of Baboon; (**g**) Pepper original image; (**h**) ciphered image of Pepper; (**i**) decrypted image of Pepper.

**Figure 10 entropy-24-01270-f010:**
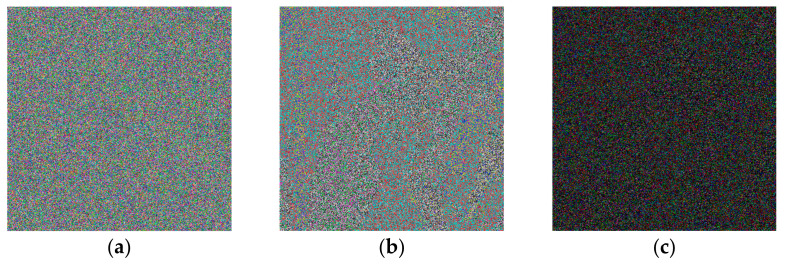
Encrypted image: (**a**) Lena cipher image; (**b**) on changing the initial value to encrypt the ciphertext; (**c**) differential image of (**a**,**b**).

**Figure 11 entropy-24-01270-f011:**
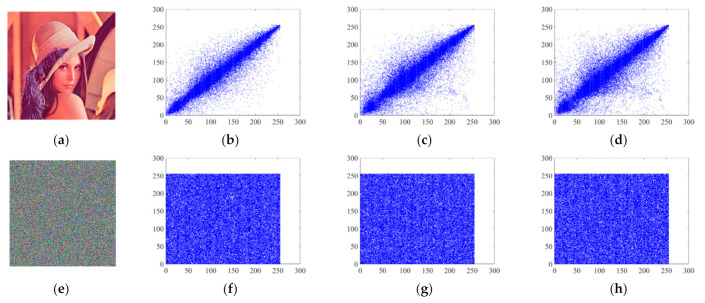
Correlations between plaintext and ciphertext in each direction: (**a**) plaintext image; (**b**) horizontal pixel correlation of plaintext image; (**c**) vertical pixel correlation of plaintext image; (**d**) diagonal pixel correlation of plaintext image; (**e**) ciphertext image; (**f**) horizontal pixel correlation of ciphertext image; (**g**) vertical pixel correlation of ciphertext image; (**h**) diagonal pixel correlation of ciphertext image.

**Figure 12 entropy-24-01270-f012:**
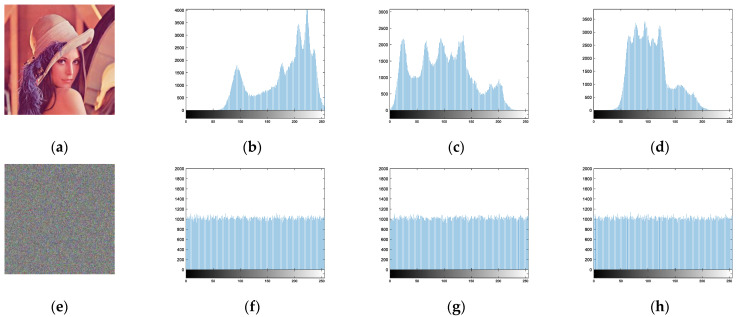
RGB image histogram of plaintext and ciphertext: (**a**) plaintext image; (**b**) plaintext R-component histogram; (**c**) plaintext G-component histogram; (**d**) plaintext B-component histogram; (**e**) ciphertext image; (**f**) ciphertext R-component histogram; (**g**) ciphertext G-component histogram; (**h**) ciphertext B-component histogram.

**Figure 13 entropy-24-01270-f013:**
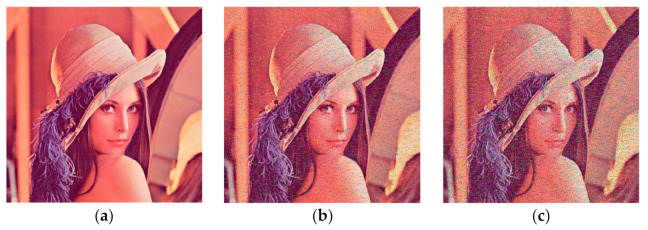
Decryption results for encrypted image: (**a**) without adding noise; (**b**) with 0.05 density noise; (**c**) with 0.2 density noise.

**Figure 14 entropy-24-01270-f014:**
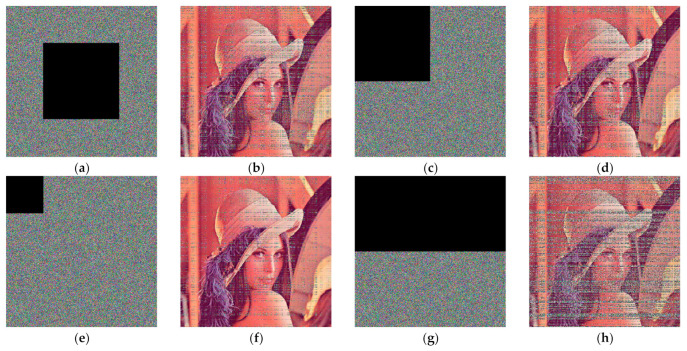
Different levels of crop attacks on different positions of cipher text images: (**a**) ciphertext crop 1/4; (**b**) decrypted image of (**a**); (**c**) ciphertext crop 1/4; (**d**) decrypted image of (**c**); (**e**) ciphertext crop 1/16; (**f**) decrypted image of (**e**); (**g**) ciphertext crop 1/2; (**h**) decrypted image of (**g**).

**Table 1 entropy-24-01270-t001:** Results of the NIST test for the quantized results s(i).

Test	*p*-Value	Result
Approximate Entropy	0.946734	Pass
Block Frequency	0.586564	Pass
Cumulative Sum 1	0.133011	Pass
Cumulative Sum 2	0.137823	Pass
FFT	0.652959	Pass
Frequency	0.818092	Pass
Linear Complexity	0.878124	Pass
Longest Run	0.213469	Pass
Nonoverlapping Template	0.238582	Pass
Overlapping Template	0.429132	Pass
Random Excursion	0.245937	Pass
Random Excursions Variant	0.314578	Pass
Rank	0.866239	Pass
Runs	0.352398	Pass
Serial 1	0.190145	Pass
Serial 2	0.306519	Pass
Universal	0.114032	Pass

**Table 2 entropy-24-01270-t002:** DNA encoding rules.

Rule	1	2	3	4	5	6	7	8
00	A	A	T	T	C	C	G	G
11	T	T	A	A	G	G	C	C
01	C	G	C	G	A	T	A	T
10	G	C	G	C	T	A	T	A

**Table 3 entropy-24-01270-t003:** DNA addition rules.

+	A	T	G	C
A	A	T	G	C
T	T	G	C	A
G	G	C	A	T
C	C	A	T	G

**Table 4 entropy-24-01270-t004:** DNA subtraction rules.

−	A	T	G	C
A	A	C	G	T
T	T	A	C	G
G	G	T	A	C
C	C	G	T	A

**Table 5 entropy-24-01270-t005:** DNA XOR rules.

XOR	A	T	G	C
A	A	T	G	C
T	T	A	C	G
G	G	C	A	T
C	C	G	T	A

**Table 6 entropy-24-01270-t006:** RGB Correlation coefficients.

Correlation Coefficients		Original Image	Ciphered Image
Horizontal	R	0.9625	−0.0020
	G	0.9798	0.0012
	B	0.9650	0.0002
Vertical	R	0.9691	−0.0010
	G	0.9832	−0.0043
	B	0.9609	0.0014
Diagonal	R	0.9573	−0.0044
	G	0.9675	0.0051
	B	0.9411	−0.0013

**Table 7 entropy-24-01270-t007:** Information entropy.

Algorithm	Information Entropy			
	R	G	B	Mean
Our scheme	7.9976	7.9974	7.9975	7.9975
Ref. [29]	7.9967	7.9973	7.9970	7.9970
Ref. [30]	7.9974	7.9962	7.9972	7.9969
Ref. [31]	7.9973	7.9969	7.9971	7.9971
Ref. [32]	7.9974	7.9974	7.9974	7.9974
Ref. [33]	7.9975	7.9972	7.9977	7.9975
Ref. [34]	7.9970	7.9972	7.9967	7.9970

**Table 8 entropy-24-01270-t008:** Different images of NPCR and UACI.

Image	NPCR (%)	UACI (%)
Lena	99.6084	33.4513
Baboon	99.6133	33.4730
Pepper	99.6108	33.4473

**Table 9 entropy-24-01270-t009:** Comparison of NPCR and UACI.

Algorithms	NPCR (%)	UACI (%)
Our scheme (Lena)	99.6084	33.4513
Ref. [35]	99.6124	33.4438
Ref. [36]	99.6300	33.5200
Ref. [37]	99.6206	30.5300
Ref. [38]	99.5789	33.4549
Ref. [39]	99.6095	33.4705
Ref. [40]	99.7570	33.1200
Ref. [41]	99.6200	33.5700

## Data Availability

Not applicable.

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
