# Peer review of "A Novel Color Image Encryption Algorithm Based on 5-D Hyperchaotic System and DNA Sequence"

_entropy, 2022, doi:10.3390/e24091270_

Round 1
Reviewer 1 Report
The submitted manuscript is proposing a method based on a 5-D hyperchaotic system. This system has a large Lyapunov exponent. The method seems well-posed, and the given results interesting. Moreover, there are circuit simulations of chaos also proposed. I would like to suggest to add in the discussion, the reason why a so large Lyapun exponent is required in the method. If the value of this exponent is reduced, how is the method impacted? In the discussion, it is told "a new 5-D hyperchaotic system is proposed, which has not only a large Lyapunov exponent but also other good properties"; please discuss shortly the good properties.
Author Response
Please see attachment.a

Reviewer 2 Report
The paper is well written, is clear and can be well understood. It is relevant and can be of help and interest for the information security research community.
The topic is suitable for the journal, of broad international interest, significant and novel. The paper is clearly presented. The references are relevant, up to date, accessible, adequate and complete. In this paper, a new 5-D hyperchaotic system is proposed, which has good properties, including a combination of bit-level permutation and DNA sequences were used to encrypt the color images. The generated cryptographic images have been tested in various security tests and the experimental results have proved the excellent performance of the algorithm proposed in this paper. The presented experimental results can be proved that the original image and the decrypted image have no distortion and data loss; likewise, the cipher image has lost all features of the plaintext image, thus, the experimental results prove that the algorithm has good security. The paper present three examples of chosen-plaintext attack of two images whose results show the fitness and good behavior of the proposed chaotic image encryption algorithm.
The work is enough relevant to be published by the journal Entropy, because the interest of the chaotic information encryption methods is very high for the security research environment.
Reviewer 3 Report
In this manuscript, an image encryption algorithm is presented. With the exception of some awkward sentences and grammatical errors, the paper is relatively well written and technically sound, though the related work section provides a deficient analysis of existing multimedia security approaches. Among the missing references:
[1] A robust block-based image watermarking scheme using fast Hadamard transform and singular value decomposition; Proc. International Conference on Pattern Recognition, 2006.
[2] Spectral graph-theoretic approach to 3D mesh watermarking; Proceedings of Graphics Interface, 2007.
[3] Watermarking 3D models using spectral mesh compression; Signal, image and video processing, 2009.
[4] Video watermarking using wavelet transform and tensor algebra; Signal, Image and Video Processing, 2010.
[5] Dynamic rounds chaotic block cipher based on keyword abstract extraction; Entropy, 2018.
My general comments may be summarized as follows:
* The structure of the manuscript can be improved. For instance, Section 1 can be divided into two separate sections: Introduction and Related Work.
* For the sake of clarity, vectors and matrices should be denoted by boldface lower- and upper-case letters, respectively.
* The main contribution(s) in this manuscript should be clearly outlined up front in the introduction.
* The proposed algorithm depends on several parameters that need to be fine-tuned. So it is unclear how the choice of such parameters would affect the overall performance of the proposed framework.
Round 2
Reviewer 1 Report
The authors have properly answered my questions and revised the manuscript accordingyly. Thank you.